# Omentoplasty for Cervical Lymphocele after Aortic Arch Replacement

**DOI:** 10.3390/jcm13164737

**Published:** 2024-08-12

**Authors:** Nora Hertel, Khaled Dastagir, Moritz Schmelzle, Linda Feldbrügge, Florian Helms, Peter M. Vogt, Arjang Ruhparwar, Aron-Frederik Popov

**Affiliations:** 1Division of Cardiac Surgery, Department of Cardiothoracic-, Transplantation- and Vascular Surgery, Hannover Medical School, 30625 Hannover, Germany; hertel.nora@mh-hannover.de (N.H.);; 2Department of Plastic, Aesthetic, Hand and Reconstructive Surgery, Hannover Medical School, 30625 Hannover, Germany; 3Department of General and Visceral Surgery, Hannover Medical School, 30625 Hannover, Germany

**Keywords:** lymphocele, lymph fistula, omentoplasty, aortic arch replacement, cardiac surgery

## Abstract

Lymphocele formation is a rare complication after surgical procedures involving the mediastinum. While uncomplicated lymphoceles show high rates of spontaneous closure and are usually treated conservatively, surgical treatment might be required in cases with persistent or recurrent lymphoceles. We present the case of a 53-year-old male with reoccurring cervical swelling after two surgeries of the thoracic aorta. After 1.5 years, the swelling occurred for the first time and appeared for the next 2 years repeatedly without clinical or laboratory signs of infection. A cervical lymphocele was suspected, and the decision for surgical revision was made. Fibrin glue was applied to the potential leakage of the thoracic duct, and the cavity was filled with a free omental flap. This resulted in a complete regression of the swelling.

## 1. Introduction

Chylous leakage is a common complication after several surgeries in areas with many lymphatic structures such as esophagectomy, lung resection, mediastinal mass resection and oncological surgery with lymphadenectomy [1,2]. Overall, 1.9% of the patients after neck dissection present with lymphatic leakage [3]. Vascular surgery sometimes lead to lymphatic leaks as well, since lymphatic channels mostly run with vessels, and the abdominal aorta is surrounded by lymph nodes [4]. Nevertheless, lymph leaks are rarely described after cardiothoracic surgery [5,6]. Mostly, the mediastinal damage of lymphatic structures presents as postoperative chylothorax [1,7]. After heart surgery, a damaged thoracic duct can also appear as chylopericardium [8]. The common therapy is conservative, since most leaks close spontaneously [4]. For persistent leakage, there are multiple therapeutical options such as aspirating or surgical closure [2]. To our knowledge, a lymphocele which expands up until the mandibular after aortic arch replacement has not been described in the current literature.

## 2. Case Report

We report the case of a 53-year-old male without relevant comorbidities except hypertension after aortic root reconstruction, supracommissural aorta replacement, and proximal aortic arch replacement for an acute aortic dissection in February 2020. In August 2020, an elective total aortic arch, brachiocephalic trunk, and right carotid artery replacement was performed for progredient aortic arch aneurysm formation. Afterwards, he was treated with Meropenem, Rifampicin and Linezolid due to increasing leukocytes and CRP. The focus remained unknown, since all microbiological tests were negative. The infections signs normalized after treatment for two weeks. Four months postoperative, the patient presented with a superficial wound infection, which was treated successfully with clindamycin. In June 2021, an indolent swelling of the right side of the neck was noted. With lack of infectious signs (no fever or reddening), we decided on a watchful waiting approach. After intermitted partial regression, the swelling reoccurred repeatedly. Initially, repeated computed tomography (CT) showed constant fluid formation around the ascending and descending aorta without signs of anastomosis insufficiencies (Figure 1). In a 18-fluorodesoxyglucose positron emission tomography with computed tomography carried out in June 2023, a slight increase in the metabolism around the proximal aortic arch prosthesis and right subclavian artery was noted. Compared to the computed tomography findings two year earlier, the periprosthetic fluid formation extended further and reached the right common carotid artery up until the submandibular gland with a shift of the larynx to the left. A diagnostic puncture revealed no signs of bacterial or viral infection or colonization. In August 2023, the swelling reoccurred (Figure 2). A CT showed an increasing fluid accumulation around the aortic root extending to the carotid bifurcation and directly under the cutis ventral of the sternocleidomastoid muscle (Figure 3). In September 2023, the swelling persisted, and an angiography was performed to rule out a suspected leakage from the right subclavian anastomosis. Consequently, an endovascular stent graft was inserted in the right subclavian artery. However, the follow-up CT revealed remittent fluid formation. Thus, a surgical approach with opening of the swelling and placement of vacuum sealed wound dressing was performed. Intraoperatively, a wound cavity extending from the cervical incision to the aortic arch prosthesis surrounded by severe adhesions were found. In the laboratory investigation, the intraoperative aspirate showed characteristics of lymphatic fluid with cholesterol measured at 124 mg/dL and triglycerides at 47 mg/dL. It was tested for chylomicrons and found to be weakly positive. With that, the fluid cavity was identified as a cervical lymphocele. For sealing of the lymphatic fluid flow, fibrin glue, containing fibrinogen and thrombin (Tisseel^®^, Baxter, Unterschleißheim, Germany), was instilled in the cavity. The extend of the remaining cavity down to the aortic arch required coverage with an omental flap. The excision of the omental flap was performed through a minimally invasive trans-umbilical laparotomy (Figure 4). The gastroepiploic artery and vein were prepared for blood supply for the flap. Subsequently, the aortic and carotid prosthesis were exposed. Also, facial arteries and veins were exposed following an end-to-end anastomosis between the gastroepiploic artery and vein and the facial artery and vein. Satisfactory perfusion of the flap was observed by its color, and it was adapted to the size of the wound cavity. With this, complete filling of the lymphocele cavity was achieved (Figure 5). The patient was nourished parenterally with intravenous nutrition (NuTRIflex Lipid plus novo^®^, Braun, Melsungen, Germany), containing carbohydrates, amino acids, fatty chains, and electrolytes, for three days, and an empiric antibiotic therapy with cefuroxime was established. The postoperative course was uncomplicated, and the patient was discharged home seven days after surgery. In the follow-up visit 3 months postoperatively, clinical and sonographic examinations showed a satisfactory postoperative result without signs of fistula formation. Also, after three months, there was no reoccurring swelling.

## 3. Discussion

The mediastinum contains numerous lymphatic structures including the plexus, lymph nodes, and the thoracic duct [6]. Many anatomic variations of the course and termination of the thoracic duct and other lymphatic vessels exist, which makes leakages difficult to avoid [9]. Further, the walls are thin and vulnerable, which makes them easy to damage due to the mobilization of surrounding structures and usage of electrocautery in various surgical procedures [4,6,9,10]. Fortunately, in most of the cases, smaller lymph vessel perforations close spontaneously [4]. A lymphatic fistula can be diagnosed by its milky white appearance and in laboratory tests (triglycerides > 100 mg/dL) [9]. In other cases, a lymphatic leak can occur as a lymphocele. It is defined as a lymphatic fluid collection in a newly formed room without epithelial walls. Normally, it occurs a few days after surgery, but there are also cases described with a delayed onset [4,9]. It can be difficult to diagnose, since the leakage can be diffuse or very small and hard to locate, and clinical findings can be unspecific and differ according to the location [11]. The appearance of the fluid is more yellowish and not as unique as the fluid of a lymphatic fistula. The laboratory results are as well not always clearly defining [9]. Imaging through computed tomography or ultrasound can support the suspicion, and a pathological examination can establish the diagnosis postoperatively [11]. In our case, the swelling occurred almost 1.5 years after the initial surgery. The fluid was clear yellow, and the laboratory results were inconclusive. But since it occurred in a non-anatomic room in close neighborhood to the thoracic duct and other lymphatic channels, we suspected a lymphocele. This was supported by the absence of infectious signs, the sterile character of the fluid, and the exclusion of other potential diagnoses. The first-line therapy for lymphatic leaks is a conservative approach with a watch-and-wait strategy due to the possible spontaneous closure of lymph fistulas. Commonly applied first-line conservative strategies contain bed rest, a modified enteral diet or even parenteral nutrition [9]. Persistent lymphocele is best addressed by aspirating the fluid and sclerosis of the lymphatic channels [2]. If the approach fails, surgical procedures such as direct sutures of the located leakage, thoracic duct ligation or embolization are options to stop the lymphatic flow [2]. Further, different materials such as biological glue may be helpful to close diffuse lymphatic leaks [12]. Lastly, the preformed cavity can be filled surgically by homologous tissue flaps to prevent the fluid from collecting or by pleurodeses in cases of chylothorax [1,10]. In our case, a conservative approach was attempted for more than two years and remained unsuccessful. Since the continuous flow of lymphatic fluid inhibited postoperative wound cavity healing, we decided to fill the room to stop the flow and prevent a prosthetic infection. Even though it failed in our case, we suggest starting with a conservative approach. If this approach fails, our sequential strategy with closing the leakages and filling of the cavity in two steps should be considered. We applied fibrin glue into the wound cavity to close the diffuse lymph leaks. Later, we harvested an omental flap, which was transplanted as a free graft into the cavity. Flap surgery has been successful in some cases after neck dissection in the past [10,13]. Additional thoracic duct ligation was planned, but the swelling stopped reappearing after the surgery, and the patient refused the procedure. We suggest follow-up examinations after 4 weeks, 3 months, 9 months and then yearly.

To our knowledge, this is the first case of a lymphocele after thoracic aortic surgery which expands up until the submandibular area. There are case reports of cervical lymphocele after neck dissection or thyroid surgery but not after thoracic aortic surgery [10]. We suspect the fluid used the method of the lowest resistance, which happens to be the surgical field resulting in a cervical swelling and consecutively preventing adhesions to establish. The non-anatomic room was held open, and the lymphatic fluid was able to flow freely into the cavity without compression of the leaks. Fibrin glue application and free omentoplasty were identified as a successful therapeutic strategy for this rare form of lymphocele formation.

## Figures and Tables

**Figure 1 jcm-13-04737-f001:**
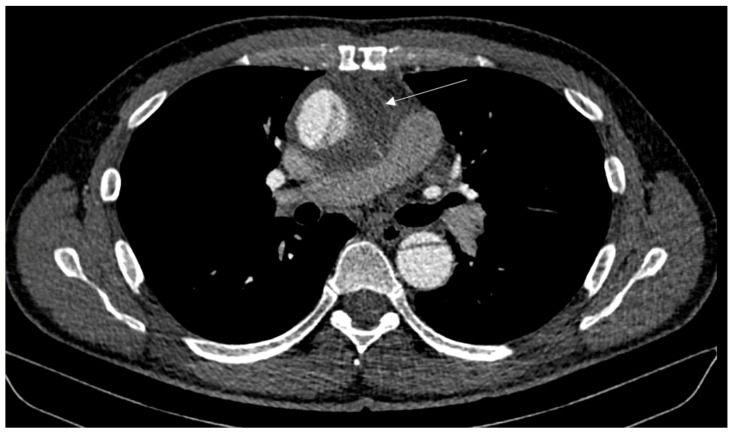
Transverse computed tomography projection one year postoperatively. The arrow shows the periprosthetic fluid formation.

**Figure 2 jcm-13-04737-f002:**
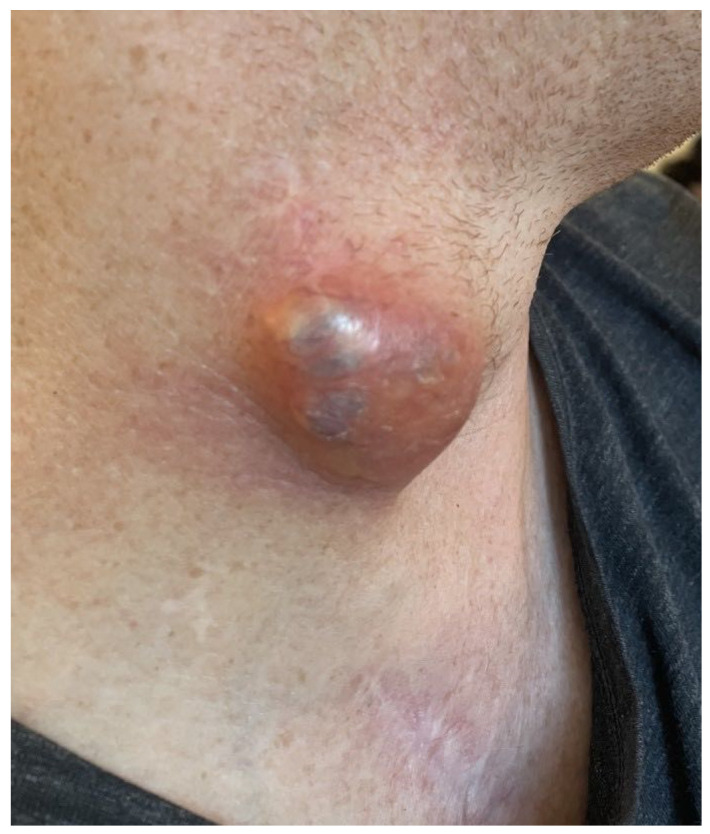
Macroscopic aspect of the indolent swelling on the right side of the neck.

**Figure 3 jcm-13-04737-f003:**
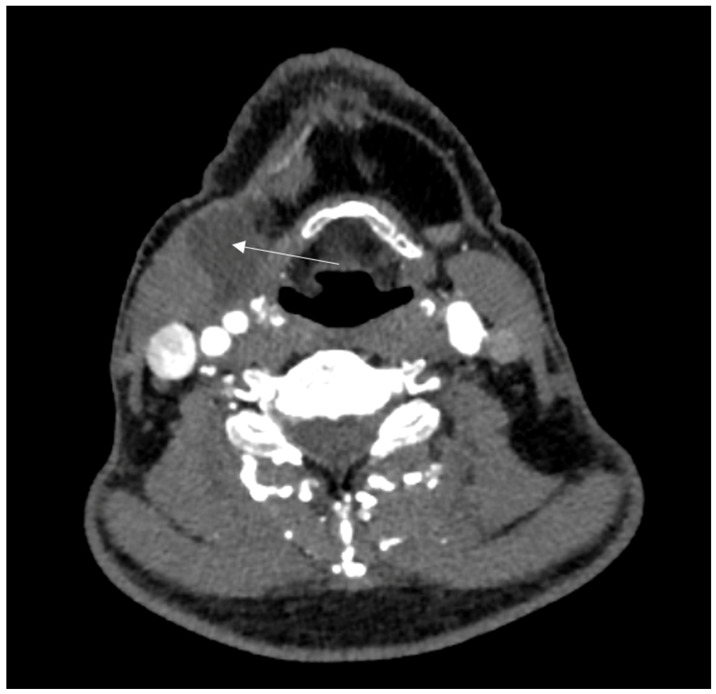
Transverse computed tomography projection two years postoperatively. The arrow shoes cervical fluid formation in a partially organized cavity.

**Figure 4 jcm-13-04737-f004:**
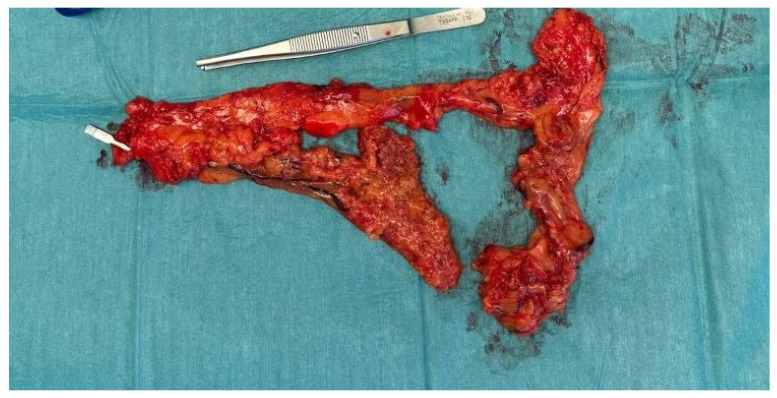
Intraoperative macroscopic aspect of the omental flap harvested in a minimally invasive fashion. The vascular clip was placed on the gastroepiploic vessel.

**Figure 5 jcm-13-04737-f005:**
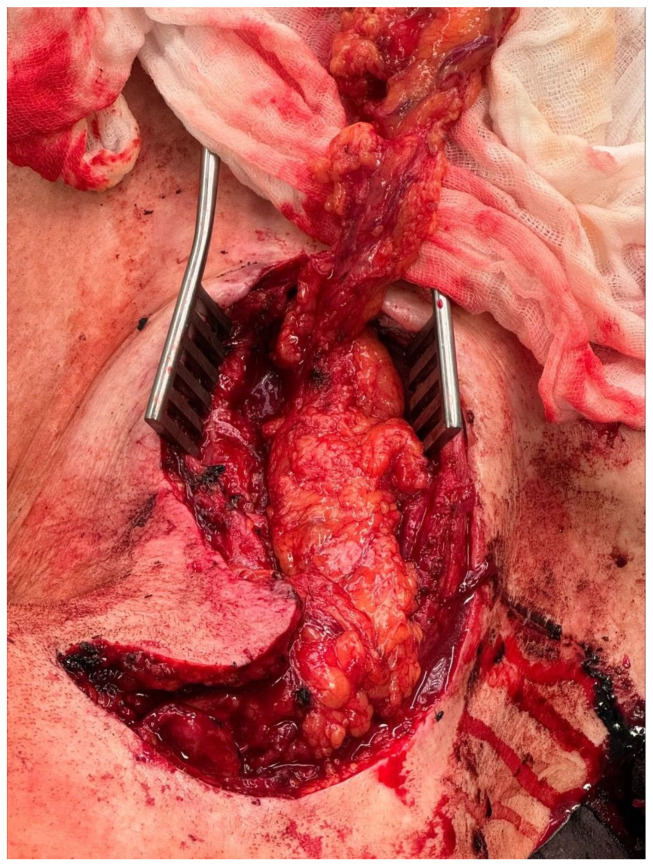
Intraoperative aspect of the omental flap placed in the cervical wound cavity after vascular anastomoses and prior to trimming. Satisfactory perfusion was achieved.

## Data Availability

The data presented in this study are available on request from the corresponding author dure to ethical reasons.

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
