# Peer review of "Omentoplasty for Cervical Lymphocele after Aortic Arch Replacement"

_jcm, 2024, doi:10.3390/jcm13164737_

Round 1

Reviewer 1 Report

Comments and Suggestions for Authors

We would like to thank the authors for submitting their manuscript to our journal. The article focuses on the surgical management through omentoplasty of cervical lymphocele, a very rare surgical complication after aortic arch replacement. The case is well described and thoroughly discussed, as well as the accompanying graphical material. The following minor revisions are required:

1) It is necessary to separate the presentation of the clinical case from the introduction. Divide the paragraph "1. Introduction" into two distinct paragraphs: Introduction and Case Presentation. In the new Introduction paragraph, briefly explain the background upon which the clinical case is based and the uniqueness of the case presented;

2) A timetable of events from the aortic arch replacement surgery, to the presentation of symptoms related to the lymphocele, to the surgical intervention, and subsequent follow-ups must be included. It would be advisable to include a graphical abstract visually depicting a timeline of events. This would help in closely following the sequence of events;

3) In the discussion, based on your experience and what can be inferred from the presentation of this case, mention what the appropriate timing for instrumental follow-up of this surgical complication might be;

4) At line 32, the authors mention the absence of signs of infection. It is necessary to provide the values of laboratory tests (complete blood count, C-reactive protein, procalcitonin, etc.);

5) Describe the cardiovascular risk factors and other comorbidities that the patient had;

6) Include the patient's pharmacological therapy and any changes that occurred in the medications taken over time.

These revisions will enhance the clarity and completeness of the article, providing a better understanding for the readers.

Author Response

Comment 1: It is necessary to separate the presentation of the clinical case from the introduction. Divide the paragraph "1. Introduction" into two distinct paragraphs: Introduction and Case Presentation. In the new Introduction paragraph, briefly explain the background upon which the clinical case is based and the uniqueness of the case presented.

Response 1: An introduction was inserted, where we explained the problem and highlighted the rareness of this complication after cardiac/aortic surgery. You can find on page 1, lines 25-38.

Comment 2: A timetable of events from the aortic arch replacement surgery to the presentation of symptoms related to the lymphocele, to the surgical intervention, and subsequent follow-ups must be included. It would be advisable to include a graphical abstract visually depicting a timeline of events. This would help in closely following the sequence of events.

Response 2: We uploaded a graphical abstract which shows the timeline of events.

Comment 3: In the discussion, based on your experience and what can be inferred from the presentation of this case, mention what the appropriate timing for instrumental follow-up of this surgical complication might be;

Response 3: On page 3 in lines 122-124 you now find, what we inferred of our case. We would still recommend to start with a conservative approach, since most lymphatic leakages are treated like this successfully. Our suggestion for follow up examinations can be found on page 3, lines 129-130. Since this was our first case of this dimensions, it is just a recommendation and might need adjustment, when there is more experience with comparable cases.

Comment 4: At line 32, the authors mention the absence of signs of infection. It is necessary to provide the values of laboratory tests (complete blood count, C-reactive protein, procalcitonin, etc.);

Response 4: Unfortunately, there are no blood parameters from the visit in June 2021. With lack of infectious signs, we referred to absent fever or reddening. It is now specified in the text (p. 2, line 50)

Comment 5: Describe the cardiovascular risk factors and other comorbidities that the patient had;

Response 5: The patient had hypertension. There were no other comorbidities or cardiovascular risk factors. The hypertension is now mentioned in the text (p. 1, line 40-41).

Comment 6: Include the patient's pharmacological therapy and any changes that occurred in the medications taken over time.

Response 6: The daily medication of the patient included only medication for hypertension and ASS. We specified the antibiotics, which were used over time (p. 2, line 45).

We thank you for your valuable notes and hope the revisions helped to enhance the clarity of our case report.

Reviewer 2 Report

Comments and Suggestions for Authors

I reviewed with interest the manuscript by Nora Hertel et al "Omentoplasty for cervical lymphocele after aortic arch replacement". The authors described a truly rare clinical case with the occurrence of cervical lymphocele after aortic arch replacement. The article presents the results of the course of this rare complication, diagnostic search and successful treatment tactics. This case may be of interest to practicing doctors.

Notes on the text of the manuscript:

1. Sources in the list of references are not placed in the order of their mention in the text.

2. Perhaps the authors should use this source in the discussion (ref. 1, see below)

3. Although the text of the manuscript begins with the heading “1. Introduction”, in fact there is no introductory information; the article immediately begins with a description of a clinical case. It is still advisable to write an introduction in the usual sense of the word. Or (which is not very good) for section 1 use the heading "Case Report"

References:

1. Ge W, Yu DC, Chen J, Shi XB, Su L, Ye Q, Ding YT. Lymphocele: a clinical analysis of 19 cases. Int J Clin Exp Med. 2015 May 15;8(5):7342-50.

Comments on the Quality of English Language

No comments

Author Response

Comment 1: Sources in the list of references are not placed in the order of their mention in the text.

Response1 : We checked our sources again and updated our bibliography. 

Comment 2: Perhaps the authors should use this source in the discussion (ref. 1, see below)

Response 2: We are grateful for this suggestion and included their findings in our case report (p. 3, lines 103-105).

Comment 3: Although the text of the manuscript begins with the heading “1. Introduction”, in fact there is no introductory information; the article immediately begins with a description of a clinical case. It is still advisable to write an introduction in the usual sense of the word. Or (which is not very good) for section 1 use the heading "Case Report"

Response 3: We absolutely agree with your comment and included an introduction (p. 1, lines 25-38).

Reviewer 3 Report

Comments and Suggestions for Authors

The introduction section per se is lack. There should be literature data about the problem

 I think that there is no need to describe the normal anatomy of the lymphatic structures in details. A short mention is enough

 What kind of fibrin glue was used?

 How surgeons assessed satisfactory perfusion of the flap?

 What parenteral nourishing in the postoperative period contained?

 For discussion section would be more beneficial if authors discuss world experience of lymph leakage after cardiac/thoracic aortic surgery as it comprises with high risk of wound infection with hazardous sequences.

Author Response

Comment 1: The introduction section per se is lack. There should be literature data about the problem.

Response 1: We agree with you suggestion and included an introduction. You can find it on page 1, lines 25-38. Unfortunately, there is not much literature data about this problem. We included all the described cases after cardiac surgery, we found, but they mostly show as chylothorax or chylopericardium. A lymphocele after aortic arch replacement with a comparable dimension is currently not described in the literature.

Comment 2: I think that there is no need to describe the normal anatomy of the lymphatic structures in details. A short mention is enough

Response 2: The passage was shortened.

Comment 3: What kind of fibrin glue was used?

Response 3: The fibrin glue we used contains fibrinogen and thrombin (Tisseel, Baxter). We included this information on page 2, line 73.

Comment 4: How surgeons assessed satisfactory perfusion of the flap?

Response 4: The color of the flap indicated a good perfusion. We included this information in the text (p. 2, line 80).

Comment 5: What parenteral nourishing in the postoperative period contained.

Response 5: We used the NuTRIflex Lipid plus novo® from the company Braun, which is a multi-component system which contains carbohydrates as wells as amino acids, fatty chains and electrolytes. The information can now be found on page 2, line 82-84.

Comment 6: For discussion section would be more beneficial if authors discuss world experience of lymph leakage after cardiac/thoracic aortic surgery as it comprises with high risk of wound infection with hazardous sequences.

Response 6: Since this is a rare case and there are only few experiences of lymph leakage after aortic surgery, a discussion of the would experience is difficult due to lack of data. We included the described cases after cardiac surgery, but they mostly presented as chylothorax or chylopericardium, which are different from our case.

Round 2

Reviewer 2 Report

Comments and Suggestions for Authors

The authors answered my questions and made corrections to the text. I have no other comments.

Comments on the Quality of English Language

No comments